# Multi-Scale Imaging of the Dynamic Organization of Chromatin

**DOI:** 10.3390/ijms242115975

**Published:** 2023-11-04

**Authors:** Fabiola García Fernández, Sébastien Huet, Judith Miné-Hattab

**Affiliations:** 1Laboratory of Computational and Quantitative Biology, CNRS, Institut de Biologie Paris-Seine, Sorbonne Université, 75005 Paris, France; fabiola.garcia_fernandez@sorbonne-universite.fr; 2Univ Rennes, CNRS, IGDR (Institut de Génétique et Développement de Rennes)-UMR 6290, BIOSIT-UMS 3480, 35000 Rennes, France; sebastien.huet@univ-rennes.fr; 3Institut Universitaire de France, 75231 Paris, France

**Keywords:** chromatin organization and dynamics, high resolution imaging, DNA repair

## Abstract

Chromatin is now regarded as a heterogeneous and dynamic structure occupying a non-random position within the cell nucleus, where it plays a key role in regulating various functions of the genome. This current view of chromatin has emerged thanks to high spatiotemporal resolution imaging, among other new technologies developed in the last decade. In addition to challenging early assumptions of chromatin being regular and static, high spatiotemporal resolution imaging made it possible to visualize and characterize different chromatin structures such as clutches, domains and compartments. More specifically, super-resolution microscopy facilitates the study of different cellular processes at a nucleosome scale, providing a multi-scale view of chromatin behavior within the nucleus in different environments. In this review, we describe recent imaging techniques to study the dynamic organization of chromatin at high spatiotemporal resolution. We also discuss recent findings, elucidated by these techniques, on the chromatin landscape during different cellular processes, with an emphasis on the DNA damage response.

## 1. Introduction

Over the last two decades, extensive studies across different model systems have revealed that nuclear organization plays fundamental biological roles. Chromosomes, and the genes they host, are arranged within the three-dimensional space of the nucleus in a specific manner, occupying a preferred location. Far from being a polymer with a static organization, chromatin diffuses inside living cells with specific properties, and its dynamics are often altered following specific stresses or in cells from diseased tissue. Several key questions remain open. How does our genome fit into a nucleus around 200,000 times smaller than unwrapped DNA? To what extent does the chromatin architecture in topological regions contribute to nuclear function? On what length scales does this structural organization occur? How is chromatin dynamically organized during the cell cycle? How do essential nuclear functions such as DNA replication, transcription or repair alter the dynamic organization of chromatin?

With the rapid development of high spatiotemporal resolution imaging techniques during the last 15 years, it became possible to visualize the structure and dynamics of chromatin at the scale of a few nanometers in space and milliseconds in time. These observations revealed different levels of chromatin organization and dynamics. In this review, we will first present single imaging techniques allowing for the study of dynamic organization of chromatin at unprecedented resolution. Then, we will describe the state of the art on the multi-scale organization of chromatin. Finally, we will present how several biological processes modify the dynamic organization of chromatin, focusing on DNA damage.

## 2. Recent Fluorescence-Based Techniques to Study the Dynamic Organization of Chromatin at High Spatiotemporal Resolution

The resolution in optical microscopy is limited to 0.6 L/NA ~250 nm (where l is the wavelength used to observe the sample and NA the numerical aperture of the objective). As a consequence, each fluorescent molecule appears as a spot of ~250 nm in radius, and spots closer than this distance cannot be resolved by classical microscopy. During the last 20 years, several techniques emerged to break the diffraction limit and observe structures smaller than 250 nm [1,2]. The aim of this section is to highlight high spatiotemporal resolution imaging techniques that have been used to address major questions on chromatin organization and dynamics. In particular, we will describe FRET-FLIM, FCS, single molecule localization microscopy and multiplexed FISH combined with super-resolution imaging.

### 2.1. FRET-FLIM

FRET-FLIM (Förster resonance energy transfer by fluorescence lifetime imaging) is the method of choice to measure the interactions between proteins in living cells. Fluorescence resonance energy transfer (FRET) involves the energy transfer through dipole–dipole coupling of a donor and acceptor chromophore. This transfer requires that the fluorescence emission spectra of the donor overlap with the absorption spectra of the acceptor molecule and that the distance between the two fluorophores is within a few nanometers. Since this ranges with the typical size of protein complexes, FRET is an ideal tool for the study of protein–protein interactions. Several methods have been developed to assess FRET efficiency, probably the most quantitative being the measurement of the fluorescence lifetime of the donor (FLIM), which has been used to measure the spatiotemporal dynamics of chromatin in living cells [3,4,5].

### 2.2. FCS

Fluorescence correlation spectroscopy (FCS) is based on the analysis of fluorescence fluctuations arising from fluorescently tagged proteins moving in and out of the constant laser beam of a confocal setup [6,7]. By calculating the autocorrelation of these fluctuations, it is possible to estimate the characteristic time spent by the tagged molecules within the focal volume as well as their average number within this volume. Further quantitative analysis of the autocorrelation curves can also reveal the mode of diffusion (Brownian motion, sub-diffusion, etc.) as well as the existence of sub-populations of molecules [8,9]. Applied to nuclear proteins, FCS provides information on the binding of proteins to chromatin and on their mobility within the chromosomal environment at the microsecond time scale [10]. For example, Wachsmuth et al. assessed the local movement of the chromatin fiber by analyzing fluorescence fluctuations arising from the linker histone variant H1.0 tagged with EGFP [11]. Similarly, Bancaud et al. showed how photoactivatable GFP dimers can easily diffuse throughout the nucleus within seconds, without dense structures such as heterochromatin not visibly slowing down their motion [12].

### 2.3. Single Molecule Localization Microscopy: PALM–STORM-SPT

Single molecule localization microscopy allows the imaging of single molecules in cells. The trick is to light only a small fraction of the fluorophores or dyes in the sample so that their relative distance is greater than 250 nm. In this case, each spot in the image corresponds to a single molecule that can be localized at high spatial resolution by determining the center of its point spread function (PSF)—a common tool to describe the response of an imaging system to a point source or object. To obtain a full image of the sample, a few molecules are lighted on, localized and photobleached: this process is repeated for each frame of a movie until the structure can be accurately reconstructed. This principle is named PALM [13] (photoactivated localization microscopy) or STORM (stochastic optical reconstruction microscopy) [14]. Both PALM and STORM need to be performed in fixed cells and allow the study of structures at up to 5 nm resolution, such as small organelles or local molecular clusters (Figure 1, top panel). Given the high spatial resolution accessible with these techniques and similar to what was reported for electron microscopy, it is important to keep in mind that some structures can be altered by the fixation process [15].

The difference between PALM and STORM is the nature of the fluorescent probes used to realize the “light on–localization–bleaching” sequence. PALM uses fluorescent probes directly fused to the protein of interest and emitting light stochastically at low frequency under an illumination at 405 nm. PALM is also the best technique to estimate the number of tagged molecules inside cells or in sub-compartments. In contrast, STORM uses inorganic dyes indirectly linked to the protein of interest (through a Halo, Snap GFP-antiGFP nanobody system, etc.). These dyes emit light stochastically almost indefinitely, limiting the estimation of the number of molecules.

PALM/STORM microscopy gives access to several observables that are not easily accessible without such a single molecule approach. From the cloud of points obtained by PALM/STORM, the simplest observable is the local density of proteins, defined as the number of neighbors in a certain radius around each detection. To further quantify the distribution of points, a common method consists in calculating the Voronoi diagram of a cloud of points. This diagram is tiling of the plane into regions from a discrete set of points called seeds: Each region encloses a single germ and forms the set of points in the plane closer to this germ than any other. The Delaunay triangulation, also used for PALM/STORM analysis, is dual to the Voronoi diagram set. This computation geometry analysis allows then for quantifying clusters, to access the amount of space unoccupied by the protein of interest also called the lacunarity, and to characterize these wholes (number, distribution of sizes surfaces, volumes and shapes) [16,17].

The dynamic motion at a single molecular level can be tracked in living cells using single particle tracking (SPT): in that case, the same type of fluorescent probes can be used allowing the imaging of individual molecules one after another. However, instead of bleaching them immediately after lighting them, the aim is to follow their motion in time over multiple timeframes. To track the molecule correctly, the density of visible molecules needs to be low to avoid overlapping point emitters, i.e., mis-linking. This approach allows us to directly access the dynamics of individual molecules at unprecedented resolution, giving access to their types of motion, diffusion coefficient, residence time on a substrate, etc. (Figure 1, bottom panel). Using more advanced analysis, it is also possible to test the existence of an energy potential attracting or repelling molecules within distances smaller than the diffraction limit or to distinguish between different types of macromolecular condensates [18].

### 2.4. Multiplexed FISH Combined with Super-Resolution Imaging

Despite a very high spatial resolution, the main limitation of the imaging methods described above is their inability to identify specific genomic regions. In contrast, fluorescent in situ hybridization (FISH) and in situ sequencing-based approaches have the advantage of obtaining both the spatial and genomic information of the signal in single cells [19]. A catalyst of these approaches was the development of massively parallel oligonucleotide synthesis methods to generate customized complex oligonucleotide libraries such as Oligopaint [20], which greatly facilitate the detection of multiple non-repetitive nucleic acid species. However, these classical FISH techniques did not have the genomic and spatial resolution necessary to characterize the organization of chromatin at the sub-microscopic scale.

More recently, Beliveau et al. combined FISH and STORM to access the fine structure of chromosomes at known chromosomal loci [21]. They fluorescently marked 46 epigenetically defined genomic domains in Drosophila Kc167 cells and measured the physical volume occupied by each domain. Chromosomes exhibit fascinating behaviors on length scales of ~100 nm to 1 micron. Thus, adapting FISH and STORM techniques to image a whole chromosome will considerably widen our possibilities in the field of high spatial resolution imaging of chromosomes [22,23,24]. Another approach is to combine live cell chromatin imaging and multiplexed FISH labelling, which allows first to track the movement of chromosomal loci and then to resolve their identities [25,26].

## 3. The Basics of Chromatin Organization and Dynamics Unveiled by Super-Resolution Microscopy

### 3.1. Chromatin Organization

In eukaryotic cells, genomic DNA is assembled into nucleosomes consisting of a 146 bp of DNA wrapped around an octamer of histones H2A, H2B, H3, and H4. Single nucleosomes are connected by 20 to 75 bp of linker DNA that can bind to histone H1. Together, the nucleosome and linker form repetitive structures of 160–200 bp that is referred to as the 10-nm fiber. This 10 nm fiber was early observed in in vitro studies in which purified nucleosomes formed the well-known “beads-on-a-string” structure [27,28]. A further level of structural compaction was thought to be necessary to fit 2 m of human genomic DNA into the nucleus. Thus, following the in vitro paradigm of chromosome organization, the 10 nm chromatin fiber is assumed to fold into more ordered 30 nm fibers [29]. However, this key theory has currently been challenged, as in vivo imaging experiments have failed to detect the presence of regular 30 nm fibers [30,31,32,33,34]. Instead, STORM-super-resolution imaging has evidenced that nucleosomes assembled in heterogeneous groups of varying sizes, termed “**clutches**”, which are interspersed with nucleosome-depleted regions [32]. Likewise, an elegant combination of EM tomography and a DNA labeling method (ChromEMT) showed that nucleosomes assemble into disordered chains with diameters ranging from 5 to 24 nm but at different concentration densities in the nucleus [31]. These two latter examples are in disagreement with the existence of a well-defined 30 nm fiber and highlight the significant heterogeneity and diameter variation in the nucleosome clusters.

Beyond nucleosome assembly, a number of large-scale structures have been described in mammalian cells using recent imaging, biophysical, biochemical and deep sequencing technologies [35,36,37,38,39,40]. Chromosome conformation capture (3C) techniques, useful for mapping chromatin interactions across the genome, have uncovered the existence of **chromatin loops** along the fiber that can range from tens of Kb to several 100 kb [37]. In addition to Hi-C evidence, a pioneering study using interferometric PALM (iPALM) has recently provided the first direct visualization of individual chromatin loops with unprecedented resolution [38]. iPALM showed large structural heterogeneity of chromatin loops involving several factors such as cohesin-mediated extrusion, CTCF anchoring and nucleosome stacking. The formation of these structures appears to be highly dynamic, at least once every cell cycle in cycling mammalian cells (~15–30 h), and are often linked to DNA transactions such as transcription and repair [41].

At a larger scale, loops that are characterized by unique chromatin features can coalesce into megabase-sized local chromatin interaction domains termed **topological association domains (TADs)** [42,43]. STORM and 3D-SIM technology allowed for characterization of TADs at the single-cell level, defining them as globular structures with sharp domain boundaries and strong physical segregation between neighboring domains [44,45]. Furthermore, single-cell analysis revealed that, similar to loops, TADs are also dynamically formed and can sporadically dissolve [39]. In addition to chromatin organization in loops and TADs, other high-order structures of approximatively 200 nm were described in the literature: from the early described chromonema fibers (100–200 nm) [46] to the more recent PALM-detected nucleosome domains (200 nm) [35], “slinky units” observed by EM (200–300 nm) [47] and chromatin blobs evidenced by deep PALM (~45–90-nm) [36]. Notably, these structures seem to share some of the properties of TADs.

The next level of chromatin organization consists of **compartments**. These compartments of several mega-bases of DNA are composed of numerous TADs grouped according to their loop density, nucleosome composition and functional state [39,48,49]. Two spatially polarized compartments were distinguished: a transcriptionally active compartment composed of open chromatin (compartment A) and an inactive compartment composed of closed chromatin (compartment B) [50]. Recent assays with sophisticated FISH technology provided a high spatiotemporal resolution view of chromatin loci in the A and B compartments organized in individual cells [51,52]. Although most A and B loci segregated in essentially non-overlapping spatial territories, confirming their spatially polarized organization [51], there was a subtle spatial overlap between a few A and B loci [52]. Interestingly, overlapping loci corresponded to a single or series of TADs, highlighting that TADs are the dynamic functional units inside the A/B compartments.

Finally, at the highest level of chromatin organization, fluorescence microscopy portrayed each chromosome occupying its own three-dimensional space in the nucleus, giving rise to “**chromosome territories**” [53,54]. Overall, as illustrated in Figure 2, eukaryote chromatin exhibits a dynamic and hierarchical organization ranging from the 10 nm fiber, loops, topological domains and A/B compartments to chromosomal territories.

Beyond the different chromatin structures described above, molecular condensates—micron-scale structures in eukaryotic cells that lack surrounding membranes but function to concentrate proteins and nucleic acids—are becoming increasingly important in understanding chromatin compartmentalization. Several models of condensates have been described according to their physical nature. The formation of these condensates appears to be involved in diverse processes, including RNA metabolism, ribosome biogenesis, the DNA damage response and signal transduction (reviewed in [55,56]).

### 3.2. Chromatin Dynamics

Far from being a polymer with a static organization, chromatin diffuses within the nucleus of living cells with specific properties. This dynamic behavior of chromatin seems to provide a degree of DNA accessibility and, thus, prompt nuclear scanning and possible functional interactions with molecular partners critical for DNA transaction processes such as replication, transcription and repair [36,57]. However, what is the nature of chromatin motion?

Several methods exist to analyze chromatin dynamics. A major analytical tool used to quantify chromatin loci motion consists of measuring their position (*x*, *y*, *z*) over time and calculating their mean square displacement (*MSD*) [58].
$$\mathrm{MSD}\equiv \langle {\left|x(t)-x_{0}\right|}^{2}\rangle =\frac{1}{N}\sum_{i=1}^{N}{\left|x^{\left(i\right)}(t)-x^{\left(i\right)}(0)\right|}^{2}\;MSD\left(n\cdot \Delta t\right)=\frac{1}{N-n}\overset{N-n}{\underset{i=1}{\sum }}\left[{\left(x_{i+n}-x_{i}\right)}^{2}+{\left(y_{i+n}-y_{i}\right)}^{2}\right]$$

The *MSD* averages all the displacements made by a locus over a given time step. Besides giving a quantitative readout of chromatin motion, the shape of the *MSD* curve also reveals the nature of this motion (Figure 1, bottom panel): while a linear increase in the *MSD* with the time interval indicates **Brownian** motion, an *MSD* curve deflected upwards or downwards means **super-diffusive** or **sub-diffusive** motion, respectively [59]. Interestingly, *MSD* curves generated from chromatin loci in yeast, flies and mammals revealed sub-diffusive motion [60]. Several kinds of sub-diffusive motions have been described depending on the way particles explore the nuclear space. When the motion is confined inside a sub-volume of the nucleus, the motion is called confined, and the *MSD* exhibits a plateau: $\mathrm{MSD}\left(t\right)=R_{\infty }^{2}\left(1-e^{-2dDt/R_{\infty }^{2}}\right)+4\sigma^{2}$, where d is the dimension of the motion, *D* is the diffusion coefficient, $R_{\infty }$ is the plateau, and *σ* is the noise due to experimental measurements. The confinement radius from which the particle cannot escape is given by $R_{c}=R_{\infty }\sqrt{\frac{2+d}{2}}$. When the motion is not a simple confinement but is modulated in time and space with scaling properties, the diffusion is called anomalous. In this case, sub-diffusive particles are constrained, but, unlike confined loci, they can diffuse without boundaries and thus reach further targets if given enough time. For sub-diffusive motion, the *MSD* exhibits a power law and is fitted: $\mathrm{MSD}\left(t\right)=A\;t{\;}^{\alpha }+\varepsilon$ where α, the anomalous exponent, is smaller than 1, *A* is the anomalous diffusion coefficient, and *ε* is the noise. Combining *MSD* analysis with physical modelling of chromatin described by the Rouse model, it is possible to access the properties of the environment in which chromatin is diffusing, such as the local viscosity and the chromatin persistence length. More specifically, the Rouse model predicts that *MSD*(*t*) follows a power law (*MSD* ~ *A t* 0.5) with an anomalous diffusion coefficient increasing as a function of the chromatin’s persistence length *Lp* as $A=\sqrt{\frac{64\;L_{p\;}^{2}k_{B}T}{3\pi \zeta }}$, (*k_B_T* is the Boltzmann thermal energy, and *ζ* is the monomer friction coefficient) [61]. In other words, when chromatin follows anomalous motion with an anomalous coefficient of 0.5, the amplitude of the *MSD* is directly proportional to chromatin’s stiffness and can be easily identified on a log–log scale *MSD* plot.

Although most of the studies describe chromatin mobility with one model, multi-scale tracking of chromatin in yeast reveals that a single mode of diffusion is not sufficient to describe DNA motion at different time scales. Instead, DNA motion is composed of several diffusion regimes that simultaneously drive DNA at each time scale [62]. An interpretation of such multi-scale dynamics is that the different scales of chromatin organization translate into different scales in chromatin mobility that can be independently regulated.

The tracking of individual histones by SPT has also provided new insights about chromatin movement at the nucleosome scale. Since the trajectories obtained by SPT are often short, they can be analyzed using the probability density function (PDF). Further analyses, such as distribution angles or residence time, are useful to extract additional dynamic features (Figure 1, bottom panel). For example, most SPT studies have distinguished two histone populations based on their diffusion [63,64,65,66]: a slow-population (with a D < 0.01 µm^2^/s) that constitutes more than ~70% of all histones within a nucleus and a fast population (with a D > 2 µm^2^/s) that constitutes the remaining ~30%. These two populations were interpreted as chromatin-bound and chromatin-free histones, respectively [65]. Interestingly, these two diffusion states were modeled as spatially separate domains within the nucleus, highlighting the heterogeneity of chromatin dynamics in living cells [67,68]. However, new evidence showed that individual histone H2B molecules can dynamically switch between slow and fast states, challenging the notion of spatially separated domains [63]. In line with this, some studies have unveiled a third histone population (with a D −0.5 µm^2^/s), which would represent histones that transiently interact with chromatin [69,70]. Importantly, similar to chromatin loci motion, nucleosome dynamics were also characterized as sub-diffusive, either anomalous or confined.

A question that remains to be answered is how the dynamics of individual histones can be correlated to that of chromatin loci or domains. If we compare the diffusion coefficient of a single nucleosome and of a *lacO*-labeled chromatin region (encompassing 20–50 nucleosomes), the diffusion coefficient of a single histone tracked at 0–30 ms time intervals (0.032 µm^2^/s) is approximately 100 times higher than that of a chromatin locus tracked at the same time scales for several minutes (1 × 10^−4^ µm^2^/s) [57].

While the dynamic of a chromatin locus corresponds to the diffusion of large chromatin fiber regions, histone mobility measured by SPT can capture the local diffusion of nucleosomes wrapped into chromatin, their rapid turnover and free histones.

A study using SPT-PALM has examined more precisely whether individual histone H2B tagged nucleosome’s movement reflects the dynamics of the DNA replication domain to which they belong. Interestingly, the *MSD* plot of nucleosome movement was similar to that of replication foci movement, highlighting correlated movements [35]. Likewise, it was shown that the movements of H2B histones within ∼0.3 μm^2^ chromatin microdomains were highly correlated. Interestingly, the correlation distances between these microdomains were up to 2 µm in the cell nucleus, spanning chromatin compartments and even beyond chromosomal territories [68]. Similarly, deep-PALM experiments emphasized the presence of spatiotemporal cross-correlations between chromatin structure and dynamics, extending several micrometers in space and tens of seconds in time [36]. While chromatin density loses its correlation after 3–4 µm and roughly 40 s, chromatin mobility correlations extend over ~6 µm and persist for at least 40 s [36,71].

### 3.3. Key Players in Chromatin Organization and Dynamics

The dynamic organization of chromatin is governed by several factors that are essential for its changes in compaction and thus its compartmentalization. Recent high temporal resolution and single molecule imaging studies revealed that the chromosome structural maintenance complexes (SMCs), consisting of cohesins, condensins and Smc5/6, are crucial in the extrusion of DNA loops and thus in the assembly of higher-order chromatin structures [72,73,74,75]. It has been shown by 3D SIM and Hi-C technology that TADs are unfolded in cohesion-depleted cells [45,76]. Likewise, cohesin depletion led to decondensation of the nucleosome domains detected by PALM in Nozaki et al. [35]. Strikingly, whereas TADs and domains were greatly reduced in cohesin-depleted cells, the detectability of chromosome compartments by Hi-C was increased. This is explained by an increase in long-range interactions in the absence of cohesin, which results in less contact specificity within compartments and a weakening in compartmentalization strength [77]. Consistent with an increased long-range interaction in cells lacking cohesin, higher dynamics of chromatin loci and domains were observed in the absence of cohesin [35,78,79,80,81]. However, super-resolution microscopy recently demonstrated the persistence of A/B compartments after cohesin depletion [82]. At the chromosome level, condensin is the main SMC complex that is involved in the formation of chromatin territories [83,84]. In addition to chromosome territories, a Hi-C study of genomic organization in 24 species covering the eukaryotic kingdom revealed a strong correlation between condensin and the Rabl configuration—a highly conserved chromosome conformation in a large number of eukaryotic genomes characterized by the clustering of centromeres and the anchoring of telomeres to the nuclear envelope [84].

**Figure 2 ijms-24-15975-f002:**
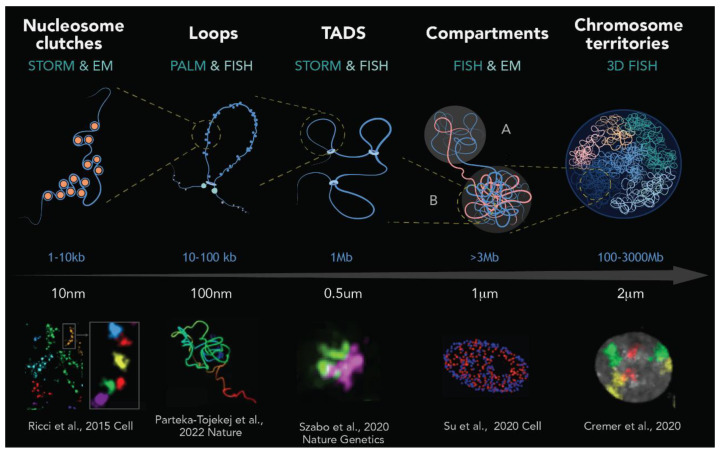
Schematic representation of genome organization in mammals. From left to right: DNA wrapped around histones forms nucleosomes, which are organized into clutches. Each nucleosome clutch contains ~1–2 kb of DNA, as revealed by super-resolution image [32]. Genomic approaches and super-resolution imaging revealed the existence of chromatin loops, which are formed by loop extrusion and in a greater extent stabilized by CTCF and the cohesin ring [38]. At the scale of ~1 Mb, chromatin loops are the base of topologically associating domains (TADs), structures with delimited boundaries and high-rate interactions inside of these domains [45]. At a higher scale, up to several mega-bases, chromatin segregates into gene-active and gene-inactive compartments (A and B, respectively) [40]. Finally, as revealed by chromosome painting and fluorescence microscopy, individual chromosomes occupy specific regions, known as chromosome territories, that are several micrometers in length [82]. Top images are representations of each level of chromatin organization (art by Olga Markova), while the bottom images are real microscopy images taken from different studies (cited at the bottom of each image).

The multiple associations of chromatin with the nuclear envelope can also shape epigenomic landscapes and high-order genome architecture (reviewed in [85]). For example, depletion of lamin A, a scaffolding protein adjacent to the inner side of the nuclear envelope, alters genome dynamics, inducing a dramatic transition from slow anomalous diffusion to fast and normal diffusion [86]. Importantly, this molecular regulation of chromatin diffusion by lamin A is considered critical for the maintenance of genome organization. Likewise, studies using species displaying Rabl configuration have shown that chromosome tethering to the nuclear envelope has a major effect on gene positioning, chromatin dynamics, transcription and DNA damage repair [87,88,89].

## 4. Chromatin Conformation as Proxy of Chromatin Accessibility

It is well established now that the chromatin packing state is uneven within the cell nucleus. Single-cell techniques such as ChromEMT and super-resolution microscopy have allowed the mapping of chromatin within a cell nucleus according to its density [31,32,36,47,90]. In these maps, two types of regions previously observed by electron microscopy and soft X-ray tomography can be differentiated: highly dense regions such as the nuclear periphery and nucleolus, known as heterochromatin (HC), and much less dense regions dispersed in the center of the nucleoplasm, known as euchromatin (EC) [91,92,93]. However, a recent study applying FLIM-FRET microscopy to analyze the spatial organization at the nanometer-range proximity between nucleosomes, termed “nanocompaction”, showed that, contrary to expectations, constitutive HC is much less compacted than bulk chromatin [3]. This suggests a new view of the distribution in nucleosomes in HC versus EC: more frequent nucleosome–nucleosome contact would occur in EC (closer than 10 nm) than in HC, but these clusters would be more spaced, resulting in a less dense distribution at a larger scale. Interestingly, this recent evidence was found strictly in living cells, suggesting a bias caused by fixation on precedent studies.

Given that chromatin compaction and chromatin accessibility are in general strongly correlated, it is not surprising that DNA processes such as transcription, replication and repair occur differently in HC than in EC. During transcription, for example, gene expression is regulated by DNA accessibility and binding of transcription factors. Thus, HC is often associated with a transcriptionally inactive or repressed state decorated by histone H3 methylation (H3K9me2/3, H3K27me3) and heterochromatin protein-1 (HP1) association. Interestingly, in vitro and in vivo studies shown that HP1 protein forms phase-separated condensates upon binding DNA, indicating that gene silencing may occur in part through sequestration of compacted chromatin in HP1 droplets [94,95]. However, the role of HP1 in the compaction, accessibility and size of HC condensates was challenged by evidence in mice arguing that the HC condensates lack a separated liquid HP1 pool, and its compaction can alternate between two digital states (compacted or uncompacted) depending on the presence of a strong transcriptional activator [96]. Likewise, a study using biophysical modeling has recently proposed that the specific affinity of HP1 for H3K9me2/3 loci promotes the formation of stable HC condensates at HP1 levels well below those observed in vitro, highlighting that the H3K9me2/3 landscape governs the HC droplet rather than HP1 itself [97].

On the other hand, EC is characterized by active gene expression, histone depletion around transcriptional start sites, histone H3 acetylated and methylated histones marks (H3K27ac, H3K4me3) [90]. Likewise, some evidence across species has shown how replication timing is controlled by chromatin architecture. Thus, the open-structured and frequently transcribed EC replicates more rapidly than HC [98,99,100,101,102]. Similarly, during cell differentiation, there is a progressive transition of chromatin compaction accompanied by repression of certain genes and HC-landmarks, a process termed heterochromatinization [103,104,105,106]. Genome-wide chromatin modification assays and super-resolution imaging have compared the profiles of undifferentiated embryonic stem cells (ESCs) with those of differentiated cells. Both mouse and human ESCs revealed widespread active chromatin domains, characterized by dimmer, sparser histone H2B domains and enrichment of H3K27ac, H3K4me3. In contrast, highly condensed histone H2B domains and HC marks such as H3K9m and H3K27 become more abundant in differentiated cells [32,35]. Importantly, alterations in this transition have critical consequences on proper differentiation during development [107].

Chromatin compaction was also closely correlated to constrained movement at different chromatin scales [35,43,68,104,108,109]. Heatmaps resulting from either single histone H2B tracking or tracking of H2B-bound chromatin domains showed a big heterogeneity in chromatin motion inside a nucleus, showing less movement within HC-rich regions than regions elsewhere in the genome [35,43,68,109]. Interestingly, despite evidence showing heterogeneity in chromatin dynamics due to compaction, a recent study has described a steady-state motion of interphase chromatin, independent of changes in chromatin compaction, cell cycle and DNA replication [66]. This motion profile was discovered by tracking individual nucleosomes but also chromatin loci on a second (1 s) time scale. Beyond this time scale, steady-state motion is lost, which explains the discrepancy with previous work using longer time scales. In terms of function, the steady-state behavior of chromatin may allow cells to cope with changes in the nuclear environment in order to maintain their routine cellular functions in similar nuclear environments [66].

Thus, the organizational rearrangement of chromatin during different DNA process is not haphazard but seems to have a specific function mainly regulating chromatin accessibility and motion.

## 5. DNA Repair and Genome Stability

So far, we have reviewed the organization and dynamics of chromatin in its physiological state. In the last section, we focus on the behavior of chromatin when it undergoes damage. In fact, DNA is continuously threatened by endogenous and exogenous factors that can result in different types of lesions, double strand breaks (DSBs) being the most detrimental. A network of cellular mechanisms, named together the DNA damage response (DDR), monitor DNA lesions, guarantee faithful repair and, therefore, chromosome stability. Two of the most well-known DSB repair mechanisms are homologous recombination (HR) and non-homologous end joining (NHEJ). While HR repairs DNA breaks by copying the missing information across the lesion from an undamaged template, as from the replicated sister chromatid, NHEJ does it by ligation of the broken ends after their juxtaposition (reviewed in [110,111,112,113]).

To decipher the mechanisms underlying the DDR, several methods to induce DNA damage have been developed over the years. Methods such as ionizing radiations, crosslinking agents, radiomimetic compounds and localized UV laser micro-irradiation induce damage randomly throughout the genome. In contrast, homing endonucleases and restriction enzymes allow the induction of damage at a targeted position on the genome, which is useful for understanding local chromatin changes around the break [114].

As seen in in vitro studies, chromatin constitutes a barrier to the DDR machinery [115] and, therefore, needs to be remodeled to allow DNA accessibility. Thus, the chromatin structure undergoes dynamic changes that are crucial for the DDR progression (Figure 3). Certainly, the original architecture of a damaged chromatin domain and its near and distant environment affect signaling and repair kinetics. Thus, it is not surprising that damaged HC regions required more extensive remodeling than damaged EC ones [116,117,118]. Despite massive heterochromatin unfolding upon UV irradiation, its specific histone marks and transcriptional silencing are maintained [119]. However, recent evidence showed a significant drop in HC histone markers under oxidative stress and, on the contrary, a raise in H3K9Ac levels, suggesting a positive gene expression epigenetic profile [120].

Chromatin remodeling starts with a first step of chromatin relaxation within seconds after DNA damage [68,121,122]. Interestingly, single-molecule microscopy has shown that this relaxation effect is specific to damaged chromatin, with nuclear regions distal to the damage being more compact [121]. This first remodeling event is dependent on histone poly-ADP-ribosylation and crucial for the recruitment of downstream DDR factors such as the ATM kinase [123,124]. In a second step, ATM and other kinases mediate the phosphorylation of histone H2AX (ƴ-H2AX), which spreads over a megabase-size domain of chromatin surrounding DNA damage [125,126,127]. ATM signaling has been shown to lead to a more compacted fiber [128]. This compaction level is required for upstream signaling by facilitating the recruitment of some adaptor proteins such as 53BP1 [124]. Interestingly, chromatin compaction here was also involved in loop extrusion and TAD organization [129,130,131]. Indeed, super-resolution microscopy revealed that CTCF and cohesin, loop/TAD mediators, are juxtaposed to ƴ-H2AX foci, suggesting that the formation of these domains is governed by high-order structures [131]. The potential functions of loop extrusion around DNA damage are to amplify the DDR signaling by enhancing chromatin–protein interactions and to protect 3D genome integrity during DNA repair [132,133], although a reduction in TAD number and insulation was found under hyperosmotic stress [134]. Notably, whereas compacted chromatin boosts upstream DDR signaling, it impairs downstream repair and restoration [124]. Thus, a second step of chromatin relaxation is needed to complete repair. Although this last remodeling step has not been fully characterized, it was associated with histone SUMOylation and ubiquitination [135,136].

**Figure 3 ijms-24-15975-f003:**
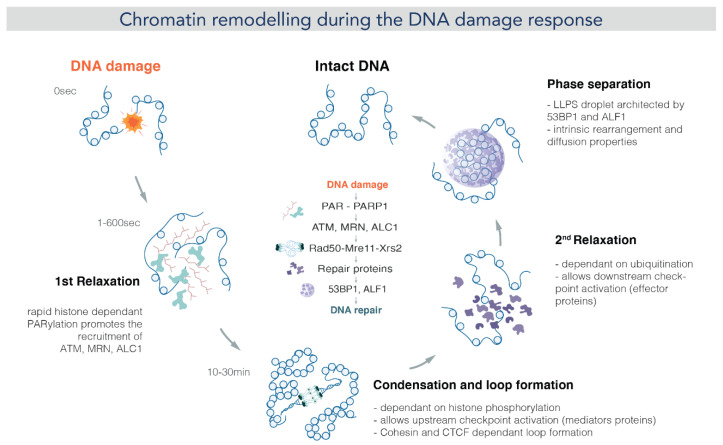
Chromatin remodeling during the DNA damage response: Upon DNA damage, PARP1 is activated and binds to the DNA damage site within seconds. Activated PARP1 catalyzes PAR chain formation and also PARylates the histone tails. This results in relaxation of the chromatin structure. Then, the PARylated PARP1 also activates and recruits other DDR factors at the DNA damage site. Among them, the ATM kinase is required to phosphorylate the H2AX histone. ATM signaling is performed through chromatin loops, which in turn triggers ƴ-H2AX spreading. The mega-base spreading of ƴ-H2AX transiently compacts chromatin around damage. This long period (10–30 min) of chromatin condensation allows the recruitment of upstream checkpoint factors. According to [124] Burgess et al., a second relaxation seems to be needed, since effector proteins cannot be recruited when chromatin is still condensed. Finally, certain repair proteins such as 53BP1 form condensates architecting chromatin in a way that repair is favored. (image by Olga Markova).

As expected, chromatin remodeling during the DDR is accompanied by changes in chromatin motion. Several studies have shown enhanced chromatin dynamics after DNA damage, which was shown to facilitate homology search during HR [118,137,138] (Figure 4). Rad51, the central protein of HR, has an essential role to promote increased mobility since in the absence of Rad51, no change is observed both at the damaged site but globally in the nucleus [137]. Miné-Hattab et al. proposed a model in which stiffening of the damaged ends by Rad51 polymerization along the single strand DNA tail, combined with globally increased stiffness, “act like a needle in a ball of yarn”, enhancing the ability of the break to traverse the chromatin meshwork. A global change in chromatin stiffening due to H2A phosphorylation has also been proposed to explain increased chromatin mobility upon DSB [139,140,141]. More recently, the direct visualization of Rad51 in living cells revealed that the dynamics of Rad51-ssDNA filaments constitute a robust search strategy, allowing DSB to rapidly explore the nuclear volume and thus enable efficient HR [142].

However, these notions come largely from studies in yeast [89,137,139,141,143,144] but remain controversial in mammals [145,146,147,148,149]. Notably, most of these studies have followed the dynamics of large regions of chromatin, while there is little evidence for the mobility of damaged chromatin at the nucleosomal level. A recent study performing histone PALM-SPT showed that motion heterogeneity changes also throughout the nucleus upon DNA damage. While heterogeneity decreases in regions around the break, the motion of regions elsewhere in the nucleus becomes homogenous [68]. This difference in the behavior of damaged and undamaged chromatin within the nucleus is proposed to be linked to the formation of repair foci or membrane-less condensates at the site of the DNA lesions. Indeed, many repair proteins relocalize from a diffuse nuclear distribution to highly concentrated nuclear foci in lesions (i.e., condensates). Consequently, these condensates change the structure of damaged chromatin, which may favor certain molecular interactions while preventing others [150]. The DDR protein 53BP1 forms one of the most studied condensates [151]. Live-cell microscopy revealed how 53BP1 condensates organize damaged chromatin into a larger repair compartment, while pushing undamaged chromatin regions away [152,153]. This is consistent with the notion of global compaction and homogenous motion in undamaged regions observed in [68,121]. Furthermore, a more recent study using STED and 3D SIM has shown that 53BP1 distribution stabilizes several neighboring loops at the break ends in an ordered circular arrangement [154]. Therefore, 53BP1 depletion disrupts this circular architecture, leading to persistent decompaction of damaged chromatin and aberrant spreading of DNA repair proteins. Indeed, inherited or acquired defects in nuclear chromatin organization during the DDR lead to genome instability. By optimizing STORM for imaging pathological tissues, a study revealed gradual chromatin decompaction and fragmentation throughout tumorigenesis. Importantly, this chromatin feature may improve diagnosis, risk stratification and cancer prevention [155].

## 6. Conclusions

During the last 10 years, high spatiotemporal resolution imaging techniques have contributed to elucidating the highly dynamic conformation of chromatin. Indeed, it is now possible to visualize and track labeled chromatin structures on size scales beyond the diffraction limit. Concomitant with the development of new single-cell techniques and computer modelling, high spatiotemporal resolution imaging allowed for better characterization of how chromatin organization regulates almost all genetic activities.

As discussed in this review, defects in chromatin conformation and dynamics have detrimental consequences in cell differentiation, gene expression and DNA damage repair. In the future, characterization of chromatin conformation in non-physiological states may provide “chromatin hallmarks” that could contribute to diagnostic imaging and thus prevent pathologies.

## Figures and Tables

**Figure 1 ijms-24-15975-f001:**
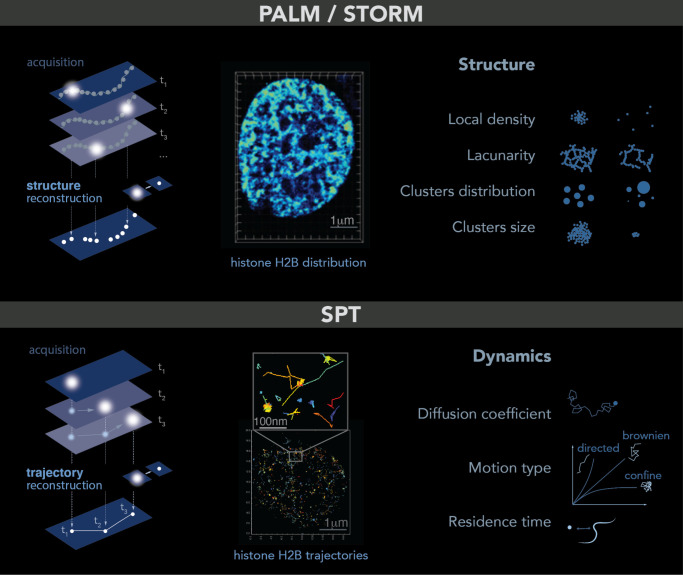
PALM and SPT: Principle, example and observables. **Top**: PALM/STORM principles. **Left**: a sparse subset of fluorescent probes is activated to produce single-particle images (represented by white circles) that do not overlap (**left**). After acquisition of images at a given time interval (t), a super-resolution image is reconstructed by plotting the measured positions of the fluorescent probes. Middle: example of histone H2B-mEOS distribution within a cell nucleus. (**Right**): further analysis of the final reconstructed images provides several parameters of the structure formed by the observed protein. Bottom: principle of SPT. (**Left**): during image acquisition, images are taken with a given exposure time (t) for the duration of several minutes. In each image, only a sparse number of emitters (white dots) are detected. Middle: using tracking and localization methods, it is possible to reconstruct the super-resolved trajectories of single molecules. Each histone H2B trajectory is represented within a cell nucleus by a colored trace. Right: different dynamic parameters can be extracted from SPT data by using mathematical approaches. (Image by Olga Markova).

**Figure 4 ijms-24-15975-f004:**
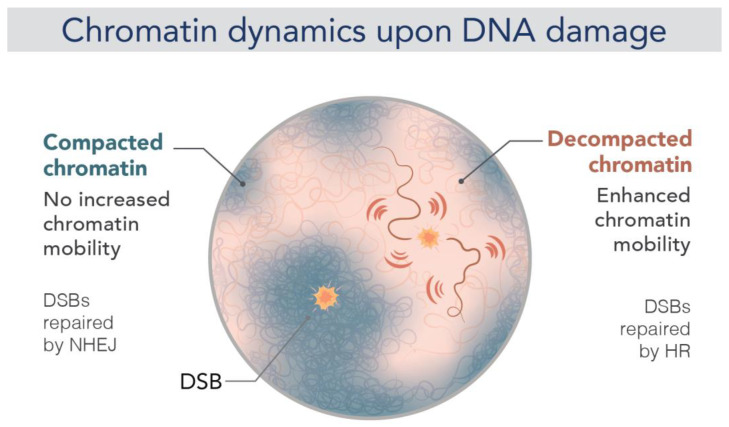
Chromatin dynamics upon DNA damage. Representation of chromatin mobility after a single DSB in a mammalian nucleus under two different conditions. When a DSB is repaired by NHEJ (**left**, blue fiber), there is no change in chromatin mobility, and it thus remains compact. Repair by HR (**right**, red fiber), on the other hand, requires increased chromatin mobility (red flash), enabling a homology search within the nucleus. This increased mobility appears to be accompanied by chromatin decompaction and stiffening (image by Olga Markova).

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
