# Peer review of "Multi-Scale Imaging of the Dynamic Organization of Chromatin"

_ijms, 2023, doi:10.3390/ijms242115975_

Round 1
Reviewer 1 Report
Comments and Suggestions for Authors
This manuscript reviews various imaging techniques for studying dynamic organization of chromatin within the nucleus during biological processes with focus on DNA damage repair. This review article is well written, covering comprehensive literatures. I have no serious concerns, but only have some minor comments to improve the manuscript.
Minor comments
1. Line 89: PALM stands for Photoactivated Localization Microscopy in Betzig et al., 2006. So change “Photo Activable” to “Photoactivated”. Also “Betzig Eric et al.” to “Betzig et al.”.
2. Lines 101, 106, 296, 312, and 360: “H2B” should be “histone H2B”.
3. Figure 1 legend: Change “Top” and “Bottom” to A and B because they are cited Figure 1A and Figure 1B in the text. In lines 101 and 106, H2B should be replaced by histone H2B.
4. Figure 1A: In middle panel, H2b should be “histone H2B” (capital B). In right panel, “several parameters of the structure”, i.e., local density, laucunarity, cluster distribution and cluster size are not explained in the text.
5. Figure 1B: In middle panel, “histone H2B”.
6. Line 157: For citation of “beads-on-a-string” structure, it would be nice to add the following: Thoma F, Koller T. Influence of histone H1 on chromatin structure. Cell. 1977. 12(1):101-7. doi: 10.1016/0092-8674(77)90188-x.
7. Line 163 citation: A mini-review “Joti et al., 2012” may be better to be replaced by its original article: Nishino Y, Eltsov M, Joti Y, Ito K, Takata H, Takahashi Y, Hihara S, Frangakis AS, Imamoto N, Ishikawa T, Maeshima K. Human mitotic chromosomes consist predominantly of irregularly folded nucleosome fibres without a 30-nm chromatin structure. EMBO J. 2012 31(7):1644-53. doi: 10.1038/emboj.2012.35.
8. Figure 2: Leftmost on the arrow, what is “pb”? Grey letters on black are dim and difficult to see.
9. Figures 3 and 4. Title in the figure “Structural modification of chromatin during DNA DSB repair cycle” and title in the legend “Chromatin remodelling during the DNA damage response” are redundant. Remove the title in the figure and modify the title in the legend if necessary. Grey fonts are too dim and difficult to see.
10. Line 241: For a pioneering study of FROS, the following article should be cited instead of Scott-Drew & Murray: Robinett CC, Straight A, Li G, Willhelm C, Sudlow G, Murray A, Belmont AS. In vivo localization of DNA sequences and visualization of large-scale chromatin organization using lac operator/repressor recognition. J Cell Biol. 1996.135:1685-700. doi: 10.1083/jcb.135.6.1685.
11. Lines 331-332: It sounds more accurate, “highly dense regions such as the nuclear periphery and nucleolus, known as heterochromatin, and much less dense regions dispersed in the nucleoplasm, known as euchromatin.”
More trivial issues
12. Line 30: Grey fonts?
13. Lines 74 115, and 124: “…” means “, etc.”?
14. Line 257: Different font?
15. Line 262: “As” doesn’t make sense.
16. Line 269: the nature of this motion? (missing “of”)
17. Line 288: “Figure 1, bottom panel” should be Figure 1B.
18. Line 405: “led” meant “lead”?
Reviewer 2 Report
Comments and Suggestions for Authors
The manuscript entitled “Multi-scale imaging of the dynamic organization of chromatin” reviews the current methods available to study chromatin organisation in living cells and present our current knowledge on chromatin dynamics in physiological conditions during cell cycle, cell differentiation and also in case of DNA damage.
The manuscript is very well written and gives a broad overview of the field. I particularly appreciate the effort made to explain the discrepancy that could be observed between studies, for example when comparing results obtained at different time scale.
However, here are a few comments that could maybe help improving even more the manuscript :
The role of structural protein such as lamin proteins, Condensin and Cohesin on chromatin organisation could be discussed (see Bronshtein et al, 2015, Nozaki et al, 2017 …).
Although most of the work on DNA damage response was done on double strand break (DSB), it would be interesting to mention what is known in the case of other types of DNA damage such as UV lesions (see Fortuny et al, 2021 for example), or in case of oxidative stress (see Casali et al, 2023) or osmotic stress (see Amat et al, 2018).
In section 3.2 Chromatin dynamics : It could be nice explaining a bit more the relation between the MSD and the diffusion constant and maybe give a bit more perspective on these values.
Minor comments :
P6 citation fig 2 : It seems that the reference “Cremer et al, Biorxiv” is not correct. Could it be “Cremer et al, 2001” instead ?.
P9 lanes 360-362 : The part of the sentence “Heatmaps resulted from H2B single or H2B domains tracking …” is not clear. Do the authors mean “Heatmap resulting from single H2B tracking and tracking of H2B bound chromatin domains …”
The section describing FROS, ParB-parS (ANCHOR), TALES and CRISPR derived methods (p6 lane239 to p7 lane262) in section 3.2 could be moved to section 2 together with the latest imaging techniques.
A few typos should be corrected :
P2 lane 94 : “It is important to be keep” should be replaced by “It is important to keep”
P4 lane 144 : “in the field high resolution” should be corrected to “in the field of high resolution”
P7 lane 262 : The sentence starting by “As” is not finished.
P7 lane 269 : “The nature this motion” should be corrected to “the nature of this motion”
P8 lane 327 : ChromET is better known as ChromEMT
P9 lane 360 : “Heatmaps resulted…” should be changed to “heatmaps resulting…”
Comments on the Quality of English Language
No comments
Reviewer 3 Report
Comments and Suggestions for Authors
This review paper titled "Multi-scale imaging of the dynamic organization of chromatin" by Fabiola García Fernández et al., presents a comprehensive review of recent advancements in understanding chromatin organization. The authors highlighted some imaging techniques for studying the structure and the dynamic changes under different (chromatin) scales, especially focused on the process of DNA damage. However, I found some reference citations are not thorough enough to cite "critical" articles for readers to take this review paper as a good example in the "dynamic organization of chromatin" field. For example, the references for FCS have only one paper from Petra Schwille.
In addition, there are many typos and grammar mistakes that made it a bit difficult to understand the whole sentence.
Here are some bullet points from the article for authors to correct and clarify:
Line4: font
Line 11: High resolution is a bit broad now here since there is a high resolution with regard to "spatially" and "temporally". It's better to better define what "high-resolution" means.
Line 27/28/29: Since liquid-liquid phase separation (LLPS) is a topic to describe dynamic chromatin structures, I suggest citing an important paper from Larson A. (https://doi.org/10.1038/nature22822)
Line 33: How is chromatin dynamically organized during the cell cycle; (should be ended with a question mark)
Line 44: is spectroscopy considered as an imaging technique?
Line49/50: Please cite proper reference
Line 61: nanometers or nm?
Line 64: English grammar
Line 70: typo?
Line 117: Single particle tracking is for "tracking" the dynamical movement at a single molecular level, not for visualizing single molecules
Line 122: miss-linking or cross-linking
Line 124: what does ..... mean?
Line 142/143/144: English issue
Line 146/147/148: what is this paragraph for? It is not explained for readers to get enough information
Line 233: inside of the nucleus?
Line 248: CRISPR-Cas9 is a tool for manipulating genes but NOT an imaging tool
Line 257: font
Line 262: AS is for ????
Line 273-276: hard to understand
Line 280-283: why don't the authors state more details adapting from the reference paper, it'd be more valuable to read this review paper. The conclusion for the cited paper is "We propose a model in which stiffening of the damaged ends by the repair complex, combined with globally increased stiffness, act like a “needle in a ball of yarn,” enhancing the ability of the break to traverse the chromatin meshwork."
Line 285-287: English is hard to understand
Line 329-332: soft x-ray tomography also reveals the different distributions of chromatin structures: Le Gros MA, Clowney EJ, Magklara A, Yen A, Markenscoff-Papadimitriou E, Colquitt B, Myllys M, Kellis M, Lomvardas S, Larabell CA. Soft X-Ray Tomography Reveals Gradual Chromatin Compaction and Reorganization during Neurogenesis In Vivo. Cell Rep. 2016 Nov 15;17(8):2125-2136. doi: 10.1016/j.celrep.2016.10.060. PMID: 27851973; PMCID: PMC5135017.
Line 382: "So" ?
Comments on the Quality of English Language
The English language needs major improvements before publication. Too many typos and wrong grammar.
